# A Granulation Tissue Detection Model to Track Chronic Wound Healing in DM Foot Ulcers

Angela Shin-Yu Lien [1,2,†] , Chen-Yao Lai [3,†] , Jyh-Da Wei [3,*] , Hui-Mei Yang [2] , Jiun-Ting Yeh [4] and Hao-Chih Tai [5,6]

1   School of Nursing, College of Medicine, Chang Gung University, Taoyuan 333, Taiwan
2   Department of Endocrinology and Metabolism, Chang Gung Memorial Hospital Linkou Branch, Taoyuan 333, Taiwan
3   Department of Computer Science and Information Engineering, College of Engineering, Artificial Intelligence Research Center, Chang Gung University, Taoyuan 333, Taiwan
4   Department of Plastic and Reconstructive Surgery, Chang Gung Memorial Hospital Linkou Branch, Taoyuan 333, Taiwan
5   Department of Surgery, National Taiwan University Hospital, Taipei 100, Taiwan
6   Department of Surgery, College of Medicine, National Taiwan University, Taipei 100, Taiwan
*   Correspondence: jdwei@mail.cgu.edu.tw; Tel.: +886-3-2118800 (ext. 3580)
†   These authors have contributed equally to this work.

**Abstract:** Diabetes mellitus (DM) foot ulcer is a chronic wound and is highly related to the mortality and morbidity of infection, and might induce sepsis and foot amputation, especially during the isolation stage of the COVID-19 pandemic. Visual observation when changing dressings is the most common and traditional method of detecting wound healing. The formation of granulation tissues plays an important role in wound healing. In the complex pathophysiology of excess and unhealthy granulation induced by infection, oxygen supply may explain the wound healing process in DM patients with multiple complicated wounds. Thus, advanced and useful tools to observe the condition of wound healing are very important for DM patients with extremities ulcers. For this purpose, we developed an artificial intelligence (AI) detection model to identify the growth of granulation tissue of the wound bed. We recruited 100 patients to provide 219 images of wounds at different healing stages from 2 hospitals. This was performed to understand the wound images of inconsistent size, and to allow self-inspection on mobile devices, having limited computing resources. We segmented those images into $32 \times 32$ blocks and used a reduced ResNet-18 model to test them individually. Furthermore, we conducted a learning method of active learning to improve the efficiency of model training. Experimental results reveal that our model can identify the region of granulation tissue with an Intersection-over-Union (IOU) rate higher than 0.5 compared to the ground truth. Multiple cross-repetitive validations also confirm that the detection results of our model may serve as an auxiliary indicator for assessing the progress of wound healing. The preliminary findings may help to identify the granulation tissue of patients with DM foot ulcer, which may lead to better long-term home care during the COVID-19 pandemic. The current limit of our model is an IOU of about 0.6. If more actual data are available, the IOU is expected to improve. We can continue to use the currently established active learning process for subsequent training.

**Keywords:** diabetes mellitus; foot ulcer; chronic wound healing; ResNet; active learning

## 1. Introduction

Diabetic foot ulcer is a tough issue for both the clinical practitioners and the patients since it is a chronic wound, which is affected by many factors, resulting in the expansion of ulcer and infection. Further exacerbation may lead to invasive debridement for the poorly healing wound, or even amputation for necrotic tissues. This does not only affect the patient's health condition and quality of life, but also increases the inpatient length of stay,

medical expenses, and may lead to mortal sepsis. Therefore, the condition of a diabetes foot ulcer should be detected early, and minor lesions of patients' extremities should also be noticed in advance.

The most important principle of chronic wound care is to continuously follow-up the wound healing condition to reduce the risk of persistent wound infection [1]. According to the wound care treatment guideline announced by EWMA (European Wound Management), traditional chronic wound care must be treated with topical treatment by daily inspection (naked-eye observation) of the wound healing status when changing dressings, by surgeons, doctors, or nurse specialists; in other words, those who have performed surgical debridement or operation, suture, or skin graft, etc., to the patient. Therefore, the first impression of the wound will be observed through visual observation and should follow the rules of MOIST (Moisture balance, Oxygen balance, Infection control, Support treatment, and Tissue management) of the wound surface [2]. That means visual observation to the wound bed surface is the most important part of wound care. For example, if the wound is too dry, it will be treated with wound gels, and if the wound is overly moist or surrounded by fluid, it will be treated with foams, hydro-fibers, or alginates for absorption to ensure a balanced moisture level, which may facilitate the formation of granulation tissue. Additionally, a paper ruler and marker pen may be used to measure and mark down the margin of the infected areas. A cotton swab may be used to measure the depth and the condition of wound discharge. Additionally, the details of the wound bed are observed, such as blood (active bleeding, blood clot, etc.), pus (infection wound or abscess, etc.), and other tissue fluid. Finally, the surgeon will prescribe the wound dressing or ointment for wound healing, such as antibiotics ointment for the infected wound, or hyperbaric oxygen therapy (HBO) as a kind of treatment for the wound care procedure.

According to MOIST, besides observing the formation of the granulation tissues, the complex pathophysiology of excess and unhealthy granulation induced by infection, oxygen supply should also be considered since the wound healing process in DM patients with multiple, complicated wounds may be affected [3].

Due to the limitations in the traditional wound care procedures which rely on professional experiences, naked-eye assessment, and manual wound determination of necrotic, sloughing, and granulation tissue, a computational approach is highly suggested as a scientific method to identify tissue and to correctly predict the healing condition of the chronic wound [4]. Relevant studies have confirmed that the diabetic foot medical care model is very sophisticated and requires cross-disciplinary cooperation through the integration of medicine (metabolism, infectious, surgery, orthopedics), nursing, digital technology, and biomedical engineering, etc., to joint efforts. With cross-disciplinary cooperation, optimal wound management can be achieved by managing the glycemic control and wound care with the assistance of modern informatics and communication technologies [5].

Nowadays, electronic products and related communication devices, such as cellphones, wearable devices, tablet computers, consumer electronics, etc., have become very common in peoples' lives. Cellphones' built-in cameras with high resolution have begun to be used in managing chronic wounds, for example in diabetic foot care [6]. It can quickly achieve the ability of screening and judging the wound condition. Some research found practical value in the actively developing digital assessment tools. According to the research of Australian scholars, the particularity of diabetic wounds' detection apps can lead to variable reliability and validity of images captured by mobile devices. Interpretation of wound images in distal assessment does not exclude the risk of distortion. However, it is still recommended as a screening or diagnostic appliance [7].

Therefore, through the modern information and communication technologies, mobile phone camera software can act as an assessment tool to analyze the preliminary diabetic foot screening. Monitoring the early pathological changes of the ulcer wound and understanding the quality of wound care at home during the COVID-19 pandemic can help to reduce the medical cost of serious infection and prevent the outcome of amputation to achieve effective diabetes health management, which is an important issue nowadays.

According to the epidemiological statistics, more than 80% of DM foot ulcers lead to amputation [8]. Usually, through debridement of the surrounding callus, removal of slough and necrotic tissue, and scraping of the microbial membrane (biofilm) on the wound surface, the new granulation tissue can proliferate and promote wound healing [9,10]. Besides, removing the physically inappropriate shear forces of the patients' feet, choosing the suitable dressing, applying topical oxygen supply, and providing related growth factors to stimulate the growth of granulation tissues are recommended treatments for diabetic foot ulcers according to the guidelines [10].

Through retrospection, literature about wound identification systems for diabetic foot ulcers has been recorded by Wang et al., who used real-time wound image data with a deep learning algorithm [11]. A recent systematic review concluded that the results of the research in AI wound assessments and monitoring systems are useful to improve DM foot clinical care, but some of them were focused on how to recognize ischemia and infection while providing strong validation or adequate reliability in assessing ischemia [12]. They use images taken by cellphone cameras to identify wounds through image smoothing and image segmentation, especially with good validity to identify the wound size and ischemic tissues [13]. However, a knowledge gap exists in the measurement of the extent of the ulcers, the differential diagnoses of the tissues, and the estimation of wound recovery. These are important focuses in the wound healing process, especially for the growth of the granulation tissues.

In recent years, almost every country in the world has been impacted by COVID-19. COVID-19 has affected the public's lives in various ways, including their health, living environments, and social networks. Most importantly, the working burden of medical staff increased while the willingness of patients to seek medical treatment decreased, thus causing a rise in amputation rates among DM foot ulcer patients.

For this reason, this study aims to develop and test an AI detection model as a screening tool by using an active learning method to identify the growth of granulation tissue and to understand the wound healing status of the DM foot ulcer wound bed for home care patients, especially during the COVID-19 pandemic.

## 2. Materials and Methods

### 2.1. The Data Source

The database is provided by two medical image centers, Chang Gung Memorial Hospital and National Taiwan University Hospital, Taiwan. We collected 219 wound pictures taken at different healing stages from 100 DM foot ulcer patients. We invited 5 specialists, including 3 plastic surgeons and 2 surgical nurse specialists who have more than 20 years of working experience in treating DM foot ulcer wounds, to validate the included samples. If inconsistency existed among these experts, the suggestion from the most senior professor and doctor of plastic surgery was used as the conclusion.

### 2.2. Institutional Review Board Statement

All subjects submitted their informed consent for inclusion before they participated in the study. The study was conducted in accordance with the Declaration of Helsinki, and the protocol was approved by the Ethics Committee of the institutional review board from university hospitals (IRB number: 201802012B0A3 and 201912056RINA). Photographic data were collected by the patients themselves or by the nurse specialists. During the above procedures, the confidentiality of the respondents was ensured.

These images were originally captured by surgeons, medical doctors, nurse specialists, or patients themselves for recording, so there was no certain sampling standard. The resolution was between 96 and 300 dpi. The physical scale for the captured objects was 80~320 pixels/cm, and the JPEG quality was about 25~75. The comprehensiveness of data collected in this research may support the development of a self-testing DM foot analyzer for home care diabetic patients.

### 2.3. Our Detection Model

The detection tool we aim to develop not only has to deal with image data of different standards but must also operate on a basic mobile device. The execution environment is very harsh—there may be no network and only limited computing resources. Our solution is to segment the images into small blocks and then train a lightweight neural network model for detection. We chose to use the ResNet [14,15] model because it is the only one that meets these conditions, unlike complex networks such as U-Net [16,17], Mask R-CNN [18,19], DeepLab [20,21], etc. In addition, we used the active learning [22,23] training method to improve the learning efficiency.

The network architecture we used in this research is depicted in Figure 1. As Figure 1a shows, we divided the wound pictures into parts with 32 × 32 sub-images, which were then divided into the red, green, blue, and gray channels. A simplified ResNet-18 model received these four channels and categorized the input data as one of the following classes: (1) granulation tissues, (2) non-granulation tissues, and (3) non-wound areas. Notably, the first class is the main one we want to detect since the growth of granulation tissues implies a positive situation for wound healing. The second class includes infected abscess, hemorrhage, skin graft, blood clot, fascia, etc. A combination of the first two classes can indicate a wound bed. The third class consists of items such as scab, skin tissue, toenails, clothing, measuring ruler, sheets, bed frames, etc., which regularly have a smoother texture than wounds.

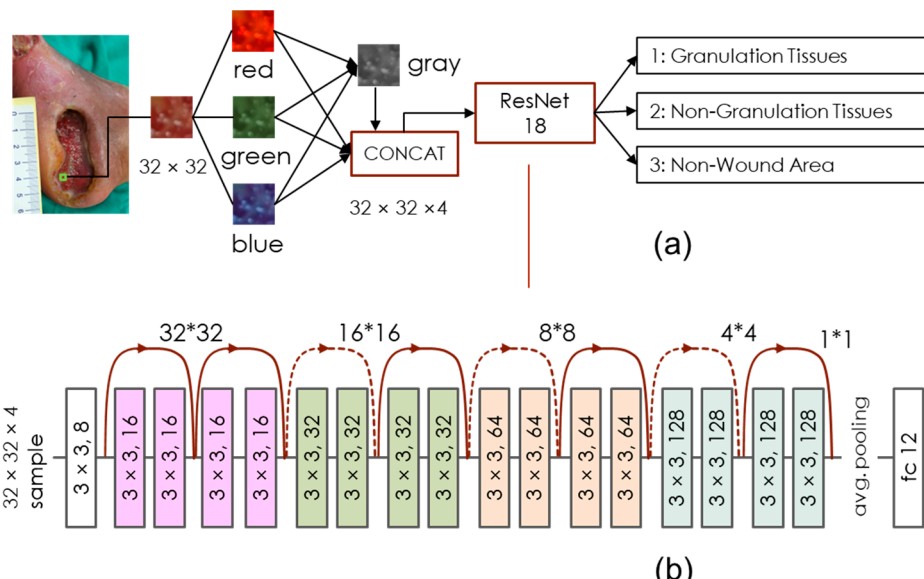

**Figure 1.** Network architecture. (**a**) This model inspects each 32 × 32 sub-image individually on the wound picture. The detection results in one of three classes: granulation tissues, non-granulation tissues, and non-wound areas. (**b**) The ResNet18 network model we used in this research is reduced to 1/4 of the original size. We can thus save the computing resources and avoid the overfitting problem.

Figure 1b shows the network components in detail. We referred to the open-source ResNet18 model [16], kept all the kernel sizes as 3 × 3, and simplified the batch sizes from 64, 128, 256, 512 to 16-32-64-128. This can reduce the network scale by 1/4 and thus save on computing resources to avoid overfitting. The output of our classification model with a 32 × 32 input block is a three-dimensional output vector generated by the soft-max activation function. For a single pixel on the test image, the classification output value $C(i,j)$ can be calculated by locating this pixel in the central 8 × 8 area of a total of 64 × 32 × 32 input blocks and summing all its output vectors, as the Equation (1) shows:

$$C(i,j) = \operatorname*{Argmax}_{k \in \{1,2,3\}} \sum_{a=0}^{7} \sum_{b=0}^{7} D(i-20+a, \; j-20+b)_k \tag{1}$$

where $D(i,j)$ is the model's three-dimensional output of the $32 \times 32$ input block, positioning the top-left corner at $(i,j)$.

### 2.4. Data Sampling and Active Learning

We developed a detection model based on examining the wound images in small patches, as this not only handles images of various sizes, but also saves computational resources. We set the block size to $32 \times 32$ after consulting five clinical specialists who were invited to label our training samples. An example of these blocks is shown in Figure 2, and we can see that the appearance of these blocks, i.e., texture, color, and gloss, most likely contains enough information for manual recognition and automatic detection.

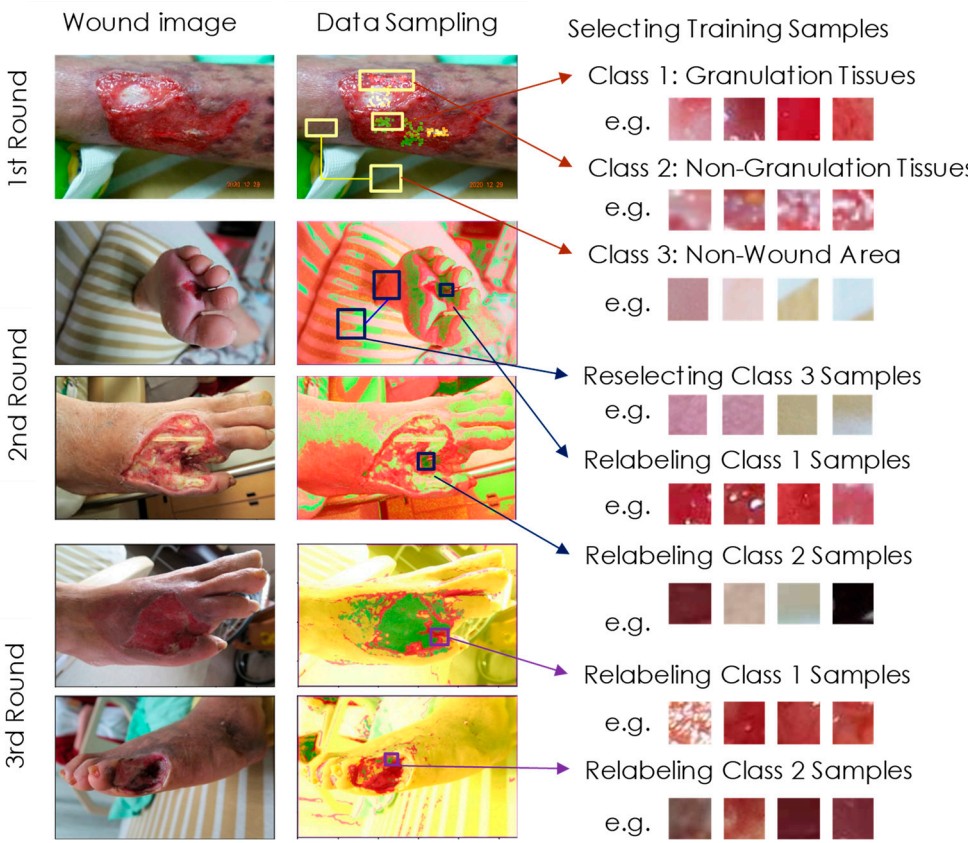

**Figure 2.** Three rounds of active learning. In the first round, we collected samples from three classes by drawing bounding boxes and generating $32 \times 32$ blocks for experts to label. We trained the model using the first set of images and applied it to detect images in the second set. When we colored the output classes with green, red, and yellow, respectively, misclassified samples could be easily relabeled and inserted into the dataset in the second round. We increased classes 1, 2, and 3 samples by 1933, 7327, and 211,397, respectively, resulting in more accurate detections before round 3. In the final round of resampling, we continued to collect misclassified samples from the third set of images. Finally, we used all samples in the dataset for model training.

We kept 9 out of 219 wound images for final testing. The remaining 210 images were divided into 3 groups for 3 rounds of the active learning process. Active learning can significantly reduce the labeling effort while improving the training efficiency. Figure 2 shows our data sampling process for active learning. We used 52 images in the first round. We then drew bounding boxes on these images, encircling the regions containing mainly granulation tissues (class 1), non-granulation tissues (class 2), and non-wound areas (class 3). For example, in a bounding box with granulation tissues as the main component, we randomly generated $32 \times 32$ blocks with their class set to class 1 by default so that clinical experts can easily label their classes. The block generation and labeling process can

be stopped at any time when we have collected enough samples in this region. We then proceeded to draw the next bounding box and label the 32 × 32 blocks generated within it.

In the first round of active learning, we obtained 1349, 1110, and 1020 samples for the 3 output classes, respectively. We then performed data augmentation by flipping and rotating these samples by 90, 180, and 270 degrees. Their intensities were also modified by gamma transformations, with $\gamma$ = 0.8, 0.9, 1.0, 1.1, and 1.2. In doing this, the data size was extended by 40 times. These samples were then used to train our detection model for 5000 epochs. After training, we applied our model to a second set of 78 wound images and started a second round of active learning. Figure 2 also shows the detection results for these images. We used three colors: green, red, and yellow, to show the classification outputs 1, 2, and 3 for each pixel computed by Equation (1), respectively. To facilitate the reader's understanding, we overlay the original image on the detection results, and set the transparency to 0.4. In the second round of data sampling, our experts quickly found misclassified regions on the image, circled them with bounding boxes, and inserted the relabeled samples into the training dataset. Therefore, at this stage, class 1 and class 2 samples were increased by 1933 and 7327, respectively. Importantly, we found that the non-wound area (class 3) almost disappeared in the output graph for this round. This can occur because class 3 should have many different types, but with relatively few examples for training. Since we could identify non-wound blocks by ourselves, we generated and added 211,397 samples to the dataset, helping this model to successfully detect the output as class 3.

The data augmentation process was then conducted as described above, and the training data used in the first round also participated in this stage. We repeated the training processes using the new dataset and started the third round of the resampling process with the remaining 80 images. As we can see in Figure 2, the non-wound area is correctly marked. Experts then focused on relabeling several misclassified regions on the output graph. The 3 types of samples increased by 25,300, 2152, and 133, respectively. Finally, we collected 28,582, 10,589, and 212,550 samples for the 3 classes (shown in Table 1). We then used all these samples and the augmented data thereof to train the final version of our detection model. The training process was repeated for 10,000 epochs, where the network model received samples from the three classes randomly but with equal probability. During the active learning process, the experts iteratively corrected the outcomes of the validation data in the first two rounds and then inserted them into the dataset as new samples. The new samples and the existing samples in the previous stage were all correct samples; therefore, they were all used to train the model in the later stage.

**Table 1.** Number of images and samples collected in the three rounds of active learning.

| | 1st Round | | 2nd Round | | 3rd Round | | Total | |
| --- | --- | --- | --- | --- | --- | --- | --- | --- |
| | #images | #samples | #images | #samples | #images | #samples | #images | #samples |
| class 1 | | 1349 | | 1933 | | 10253 | | 13535 |
| class 2 | 52 | 1011 | 78 | 7327 | 80 | 3152 | 210 | 11490 |
| class 3 | | 1020 | | 211397 | | 133 | | 212550 |

## 3. Results

As mentioned above, after three rounds of active learning, we kept nine uninvolved wound images. These images were from three patients, called Case 1 (1-1, 1-2, 1-3), Case 2 (2-1, 2-2, 2-3), and Case 3 (3-1, 3-2, 3-3). Clearly, each case went through a three-stage healing process. Of these cases, the first two had larger areas and more complex wound beds compared to case 3. The picture resolutions for these images are 150, 300, and 300 dpi, respectively, for each case. We applied our detection model to evaluate each 32 × 32 block and then used Equation (1) to identify regions of granulation tissue pixel-by-pixel. We only focused on class 1 results, as the granulation tissue growth indicated good wound healing.

Clinical specialists had also labeled the ground truth for us to verify our experimental results. Table 2 lists the IOU rates between our detection results and ground truth. For all cases, we can find that the IOU rate increased during the first, second, and third resampling, implying that active learning can efficiently improve training performance. The IOU rates after the third round of training were mostly higher than 0.5, except for case 2 and 3 which was only 0.45, and the average IOU rate reached 0.62. We also tested 10 over 52, 10 over 78, and 10 over 80 images using the well-trained model on all 3 training datasets after the third round. The images were picked randomly and had mean IOUs of 0.71, 0.67, and 0.68, respectively. Although these images have been used to train our model, the active learning process does not globally examine all image content. Therefore, the IOU rates of the training data were not significantly better than the IOU rate of the final test images. This result likely confirms the generality of our model. An IOU rate above 0.5 met the original goal of our study, revealing the ability of our detection model to aid caregivers in DM foot wound healing. Below, we use Figures 3–5 to explain the details of wound assessment for cases 1–3, respectively. Notably, the following cases provided the representative images of common DM foot ulcer wound status.

**Table 2.** Experimental results: the IOU rates compared to the groundtruth.

| Case | IOU | | |
| --- | --- | --- | --- |
| | **1st Round** | **2nd Round** | **3rd Round** |
| Case 1-1 | 0.31 | 0.57 | 0.59 |
| Case 1-2 | 0.42 | 0.61 | 0.68 |
| Case 1-3 | 0.24 | 0.51 | 0.58 |
| Case 2-1 | 0.11 | 0.5 | 0.51 |
| Case 2-2 | 0.31 | 0.51 | 0.68 |
| Case 2-3 | 0.11 | 0.39 | 0.45 |
| Case 3-1 | 0.37 | 0.61 | 0.68 |
| Case 3-2 | 0.46 | 0.65 | 0.72 |
| Case 3-3 | 0.39 | 0.62 | 0.68 |

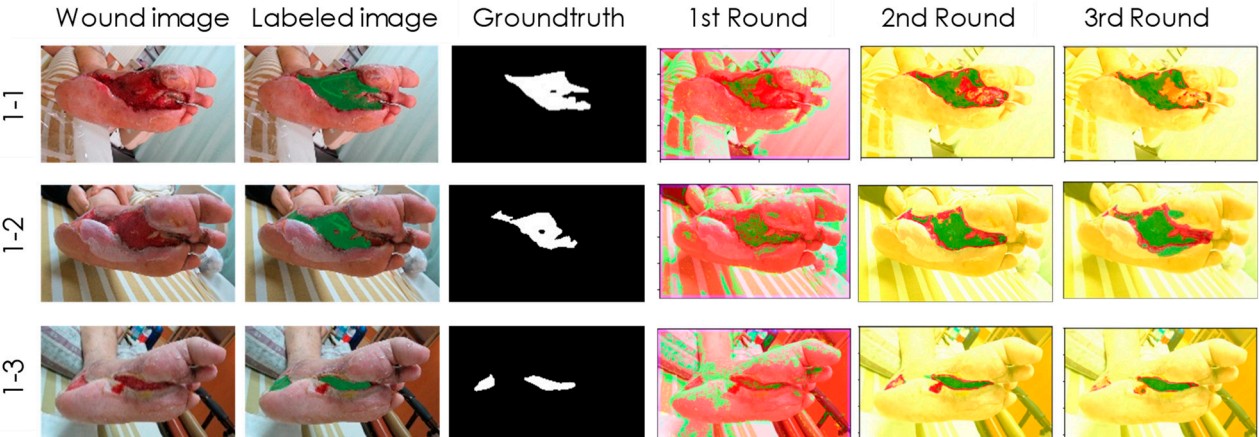

**Figure 3.** Case 1, a 65-year-old male patient. Although the IOU rate was around 0.6, the detection results precisely indicate the location and area of granulation tissues after the second and third training sessions.

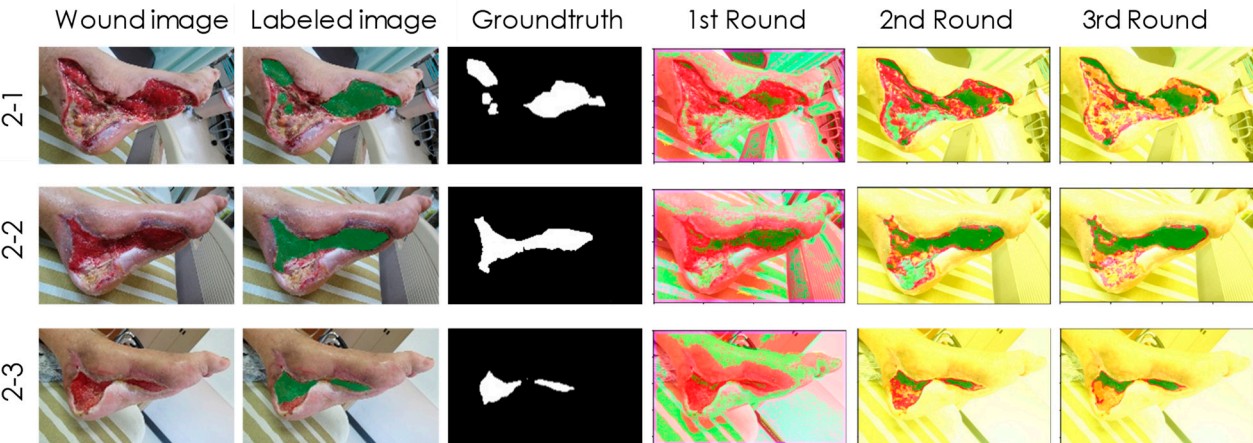

**Figure 4.** Case 2 of a 68-year-old male patient with poor DM control. The healing process was also accurately traced by our granulation detection model. However, the wound area near the heel was incorrectly classified from class 2 (non-granulation tissues) to class 3 (non-wound areas) after the third training session. This situation did not affect our detection for class 1 (granulation tissues).

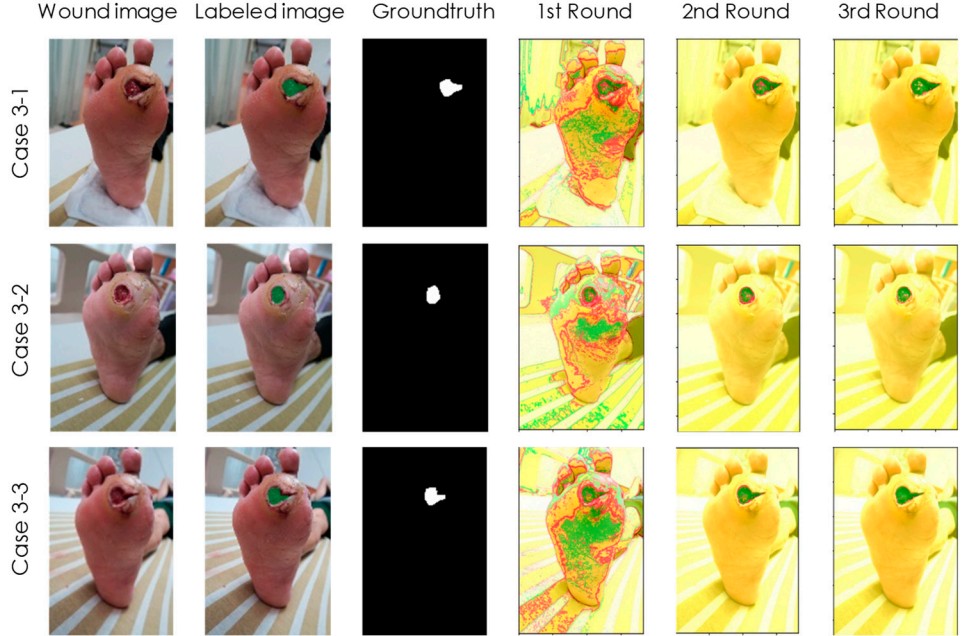

**Figure 5.** Case 3 of a 59-year-old female patient with a history of DM for over 15 years. Our detection model can accurately track the wound appearance on the surface but was unable to discover the putrescence occurring deeply behind the skin. Therefore, advance warning could not be given to prevent the occurrence of the toe amputation.

Case 1 (Figure 3) is a male patient aged 65 who has been diagnosed with Type 2 DM for 8 years. In the past 5 years, he suffered from DM neuropathy and cerebrovascular disease. He fell from his wheelchair and was injured by the pedal of the wheelchair. After debridement and wet dressing of Sulfasil cream and skin graft, the granulation tissue gradually grew hyperplastic micro-vessels, fibrous connective tissue, and many inflammatory cells. This tissue has a large network of micro-vessels, so the color appears red with slight hemorrhage, and the periphery is a new epidermis that has completed re-epithelialization. The wound was getting smaller and healing. Although the IOU rate was around 0.6, the detection results in Figure 3 precisely indicate the location and area of the granulation tissue after the second and third training sessions. Therefore, it has been affirmed by doctors and nurse specialists.

Case 2 (Figure 4) is a 68-year-old, retired truck driver with poor DM control for more than 20 years. He has been diagnosed with DM nephropathy and undergoes regular dialysis three times a week. He suffered a motorcycle injury and laceration within two weeks. The plastic surgeons debrided a large area of infected, humid wound, followed by using Sulfasil cream and New Epi (growth factor) liquid wound dressing after debridement of inflammation and necrotic tissue, and then performed skin grafting. These treatments significantly improved wound healing with granulation tissue hyperplasia. The healing process is also traced by our detection model in Figure 4. Interestingly, we found that the wound area near the heel was incorrectly classified from class 2 (non-granulation tissues) to class 3 (non-wound areas) after the third training session. However, this situation did not affect our detection for class 1 (granulation tissues).

Case 3 (Figure 5) is a 59-year-old female patient with a history of DM for over 15 years who was recently diagnosed with a poorly healing ulcer under the left toe. The wound looks small on the outside, but it is actually a deep, infected wound. After debridement, due to chronic renal insufficiency and neuropathy, osteomyelitis led to poor wound healing, which eventually led to toe amputation. In this case, our detection model accurately tracked the wound appearance on the outside, but it was unable to delve into the back of the skin, as shown in Figure 5. Therefore, early warning cannot be given to prevent unfortunate events from happening.

## 4. Discussion

As mentioned earlier, our initial goal for this study was that the IOU rate must be higher than 0.5 compared to the ground truth. Although all our experimental results, except for case 2 and 3, achieved this goal, we are curious why the IOU cannot reach 0.8, 0.9, or higher. We think there are two possible reasons. First, for some tissue samples, it can be difficult to definitively determine whether it is a granulation tissue based on their appearance. Any decision may be the correct classification. In this case, half of the ambiguous samples can be placed outside the intersection area of the IOU.

The second reason may be that the class 2 (non-granulation tissues) has various types, such as infected abscess, hemorrhage, skin graft, blood clot, fascia, etc. As long as the granulation tissue appears to be close to these types, it may be misclassified. Therefore, the conditions for the successful classification of granulation tissues are relatively strict. To confirm this argument, we set the pixels within the granulation tissues of the ground truth image to be positive and other pixels to be negative, and then calculated the positive predictive value (PPV) of the experimental results. We found that even for case 2-3 (with IOU = 0.45), its PPV value reached 0.91. Therefore, our detection results could accurately determine the location of the granulation tissues.

We also attempted to precisely detect non-granulation tissues (class 2) in our further research. By doing so, we can assess the proportion of granulation tissues in the wound bed. Here, let us discuss why we could not properly identify non-granulation tissues in case 2 and 3. As a comparison, we also review the successful detection example of case 1-2. We found that the main difference between these two cases is the resolution of the images. The image resolutions were 150 and 300 dpi for cases 1-2 and 2-3, respectively. These two images are compared in Figure 6. In Figure 6a, both granulation tissue and non-granulation tissue are clearly textured, while in Figure 6b, the texture of both is relatively flat and approximates the appearance of class 3. Therefore, in the upcoming research, we can resize all images to a consistent resolution to improve training results.

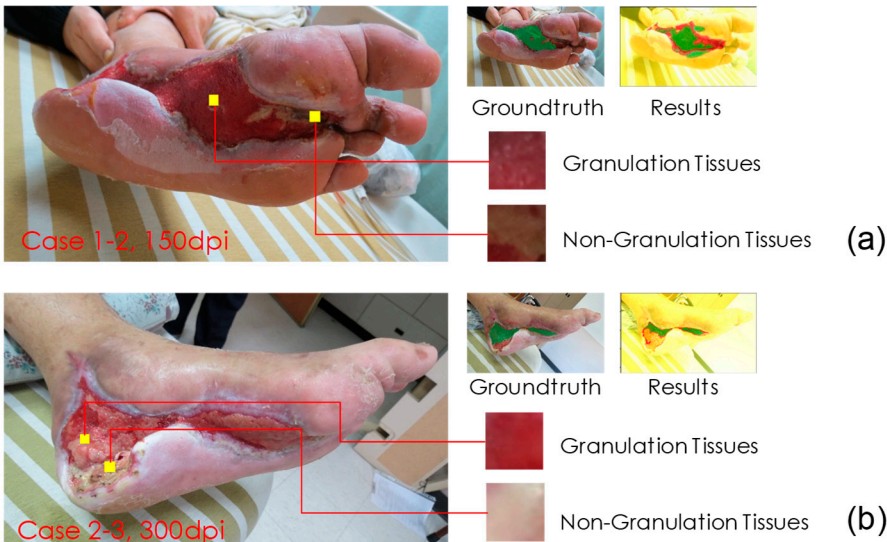

**Figure 6.** Comparison of cases 1-2 and 2-3 images. (**a**) Both granulation tissue and non-granulation tissue are clearly textured with image resolution at 150 dpi. (**b**) The texture of both is relatively flat and approximates the appearance of class 3, as the image resolution of this case is 300 dpi.

In addition to image resolution, we found other factors that contributed to misclassification. We removed images in the training data that could not be classified correctly even though they were used for training. As shown in Figure 7a, wetter tissue is more likely to reflect light when photographed, thereby affecting the detection model. Figure 7b is another example, which shows that in more complex wounds, granulation is not easily correctly detected due to mixing with other tissues. For the former, we can take several more images of different angles during detection to solve the reflection problem. For the latter, we can collect more training data—a fourth round of active learning may be performed to solve this problem.

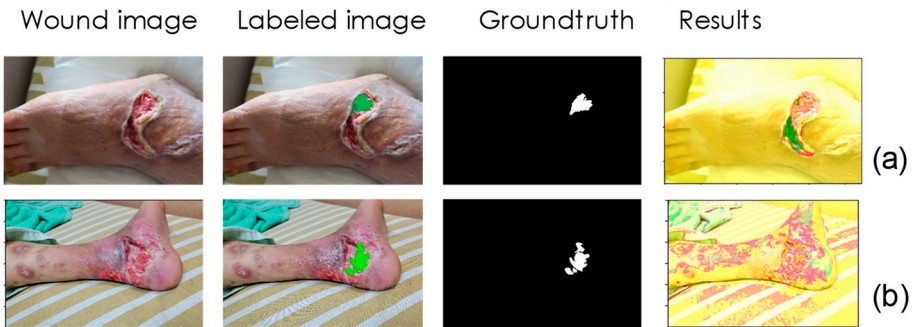

**Figure 7.** Other factors that cause misclassification. (**a**) Wetter tissue is more likely to reflect light when photographed, thereby affecting the detection model. (**b**) In more complex wounds, granulation is not easily correctly detected due to mixing with other tissues.

## 5. Conclusions

In this research, we developed and tested a granulation tissue detection model. The kernel of this model is a simplified ResNet 18 network model, which consumes relatively few computing resources and thus can perform on basic mobile devices. The input data were 32 × 32 sub-images segmented from DM wound pictures, and the classification output was divided into three classes, i.e., the granulation tissues, non-granulation tissues, and non-wound areas. We conducted a three-round active learning process to save the labeling labor and to improve training efficiency. The experimental results revealed that most of our detection results of the granulation tissues had an IOU higher than 0.5 in comparison to the ground truth. The results can thus accurately locate the place of the granulation tissues

in the wound bed. This is helpful to both the nursing staff and the DM patients' wound healing, especially during the outbreak of COVID-19.

This research constructed a mHealth device and application for patients with DM foot ulcers or chronic wounds. The present R-CNN deep machine learning model for instant diabetes foot ulcer wound imaging, for wound classification and identification, contributes quickly and sufficiently to identify the wound condition by using photo image delivery from cellphones, especially for those patients who were hesitant to visit the doctor during the COVID-19 pandemic. Furthermore, this model provides patients, healthcare professionals, and case managers a tool for the assessment of the wound status. It can be used for self-monitoring by patients, taking pictures of wounds, and uploading the pictures to the e-health cloud as a personalized medical record, as well as it can break the limitations and inequality of medical services such as time and geography. Medical staff can also use the App to view all medical data and personalized information of patients, such as blood test results and images reports, immediately communicate and discuss with each other through the App at any time, and hence increase the medical care efficiency. If there is any abnormality, the App will immediately send a warning message notification to prevent critical conditions or to decrease the possibilities of amputation induced by wound infection.

**Author Contributions:** Conceptualization, A.S.-Y.L. and J.-D.W.; data curation, A.S.-Y.L., C.-Y.L., and J.-D.W.; funding acquisition, A.S.-Y.L. and J.-D.W.; investigation, A.S.-Y.L., H.-M.Y., J.-T.Y., and H.-C.T.; methodology, A.S.-Y.L., C.-Y.L., and J.-D.W.; project administration, A.S.-Y.L. and J.-D.W.; resources, H.-M.Y., J.-T.Y., and H.-C.T.; validation, A.S.-Y.L., C.-Y.L., and J.-D.W.; writing—original draft, A.S.-Y.L. and J.-D.W.; writing—review and editing, A.S.-Y.L., C.-Y.L., and J.-D.W. All authors have read and agreed to the published version of the manuscript.

**Funding:** This research was funded by the Ministry of Science and Technology of Taiwan, grant numbers MOST-109-2314-B-182-048 and MOST-110-2221-E-182-047, and Chang Gung Medical University and Chang Gung Memorial Hospital, grant numbers BMRPDJ25 and BMRPB21.

**Institutional Review Board Statement:** The study was conducted in accordance with the Declaration of Helsinki and approved by the Institutional Review Board of Chang Gung Medical Foundation (Approval number is 201802012B0A3 and the date of approval was 2019/01/15), and National Taiwan University Hospital (Approval number is 201912056RINA and date of approval was 2021/06/01).

**Informed Consent Statement:** All subjects submitted their informed consent for inclusion before they participated in the study.

**Data Availability Statement:** The datasets used and/or analyzed during the current study are available from the Chang Gung Memorial Hospital upon reasonable request.

**Acknowledgments:** The work of the first author (A.S.-Y.L.) was supported in part by the Ministry of Science and Technology of Taiwan under the grant MOST-109-2314-B-182-048, and in part by Chang Gung University and Chang Gung Memorial Hospital, Linkou, under the grant BMRPJ25. The research of the corresponding author (J.-D.W.) was supported in part by the Ministry of Science and Technology of Taiwan under the grant MOST-110-2221-E-182-047, and in part by Chang Gung University and Chang Gung Memorial Hospital, Linkou, under the grant BMRPB21. The authors would like to express special thanks to Quanta Computer Corporation. Their Quanta Cloud Technology (QCT) high-performance servers were provided to support the computation in this study. The authors would also like to thank Yuh-Shiou Gu for assistance with the English editing of this article.

**Conflicts of Interest:** The authors declare no conflict of interest.

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
