# Peer review of "A Granulation Tissue Detection Model to Track Chronic Wound Healing in DM Foot Ulcers"

_electronics, doi:10.3390/electronics11162617_

Round 1
Reviewer 1 Report
The authors proposed an Artificial intelligence (AI) detection model to monitor diabetes mellitus foot ulcer wound recovery. The experiments were described appropriately, and the figures of the real patients were used to strongly support the results. It can be considered for publication after addressing the following major issues:
1. The abstract needs to be incorporated with the gist of the complete work in the manuscript. In the current version of the abstract, only the summary of the work is mentioned in a very broad view. Instead, please include the specific details of the research study presented in the manuscript.
2. In the abstract, please explain and provide the specifications of the scientific questions answered in the proposed study.
3. In the abstract, please include the details of the limitations of the previous research studies/technologies made in the proposed domain of study. Also, specify the scientific advancements made in the current/proposed study to overcome those limitations.
4. In the introduction, please include the current convectional granulation tissue detection method, and the advantages of this study, to guide the reader to understand the importance of the study performed. Also, include corresponding references in the text when mentioning the details.
5. In the introduction, please include the knowledge gaps existing between the current proposed study and prior studies performed in this field. Very importantly, please specify the need for the current work presented in the manuscript.
6. In the last paragraph of the introduction, kindly include the details of the broader impacts on the study made and the results achieved. It is very important to provide the future scope of the study performed to make a strong impact on the readers of the research performed/Study proposed.
7. Please explain the prospective in which the clinical specialist identifies the ground-truth granulation tissue areas? And how accurately were the areas labeled?
8. Please provide more specifications on the image processing function as it’s the key technology of the study. Also, please expand the explanation of figure 1(b) and equation 1 in detail.
9. In the manuscript it was mentioned that the specialists remove the noises in the signal. But in practical scenario if the patients are performing the self-detection, how can the patients read the results without the support of the specialists?
10. In line 305: Please clarify and provide the details to calculate the PPV value, as there are no definitions of true positive and false positive in previous sections of manuscript.
11. Please include more appropriate references from the studies that are performed more recently. In the current version of the manuscript, there are some references that are not recent publications.
12. Please revise the manuscript with English grammar. There are many places that the manuscript needs to be improved with respect to English writing.
Some minor issues:
-Line 31: “IOU” --> “Intersection-over-Union (IOU)”
-Line 82: “development” --> “develop”
-Line 89: “We received 219 wound pictures...” and line 144: “We kept nine out of 209 wound images for final testing”. Which number of the pictures is correct in this study?
-Line 149: “non-granulation tissues (category 3)” should be “(category 2)”.
-Paragraphs started from line 157 and line 175 are duplicated, please keep only one of them that the authors prefer.
-Line 224: “vilify” -->” verify”
Reviewer 2 Report
Lien et al presented a ResNet model to track chronic wound healing in diabetes foot ulcer patients. The purpose of this study was to provide a rapid tool to access the wound condition. The study is interesting. The authors presented their work very well. I only have a few comments to improve this work further.
1. The authors used 210 our of 219 images to train their model. They presented three cases from the nine unresolved cases. I suggest an IOU quantification should be presented from the training set as well, not just the validation set. Maybe an average of IOU or some other quantification from the training set is needed.
2. IOU should be spelled out in the first appearance.
3. Line 144 indicates 209 cases in total. Later the text indicates 219 cases in total. Please clarify the inconsistency. I think it should be 219.
4. It is important to include a statement of human data ethics in the methods, such as IRB approval of their institution.
Reviewer 3 Report
The manuscript presents a study devoted to development of a mobile application for chronic wound assessment using a simplified ResNet. The system is developed on a (rather small, just 219 photos) set of clinical wound images of various sizes and resolutions. Active learning has been used to train the network to recognize the granulation tissue. The topic is vital for the daily care and obtained results seem promising.
However, only three cases are presented in detail, therefore those can be considered only qualitative. The manuscript lacks a more detailed verification of the system. From those cases presented it cannot be inferred that the system is robust enough for its application.
The description of the learning phases is also misleading to some extent – if the expert corrects the outcomes of the previous learning phase (l.164), are those not-corrected also used to train the system (l.165-167, 209)?
As the dataset is rather small – was any data augmentation applied? If so – please supplement its description.
It is also quite surprising that the better resolution of the image, the texture is flatter (l.332-333).
The literature review is rather small and should be updated to include more recent publications (only 6 from the 18 references are from the last 5 years). It seems rather impossible not to find similar (machine learning or even “classical”) approaches to chronic wound assessment to compare with.
As images of real patients were used – did the experiment require ethics committee statement? If so, provide the details at the end of the manuscript.
The paper has proper structure, however suffers from editorial and language issues (listed below), therefore proofreading is highly recommended.
Specific issues:
- inconsistent/incorrect capitalization, e.g. l. 25, 68, 71, 103, 107, head of table 1 (Samples), 272, 277, 284, 338, 353 (Covid).
- unnecessary hyphenation: l. 296, 311, 318, 322, 325, 326.
- sentences hard to follow/language issues – consider rephrasing, especially: l. 41-43, 50-52, 62-64, 76-77, 127, 171-172.
- editorial: ellipsis l. 50?, not introduced acronym 3C l. 53, lacking space l. 131, l. 149 – shouldn't the granulation tissue be a category 2?, paragraphs starting from l. 157 and 175 seem to be duplicated, typo “vilify” l. 224, typo “bot” l. 352, typo in title of the reference 9.
Round 2
Reviewer 3 Report
The manuscript has been improved, however there are still language and editorial issues. Moreover there are still some statements that might be confusing to the reader. As previously, only three cases are presented in detail, therefore the outcomes can be considered only qualitative and robustness of the system for its daily application is not confirmed.
Specific issues:
- Proofreading by a native speaker is still recommended – there are still sentences hard to follow and/or language issues (e.g. l. 48, l. 58+, l. 97+, l. 101+, l. 121+, l. 157+, l. 173+, l. 208+, l. 362+).
- Inconsistent/incorrect capitalization, e.g. l. 75 (Clot), l. 120 (Biofilm), l.316 (Then This).
- Editorial: l.97 (not introduced acronym 3C), l. 163 (period instead of comma, “Photo” probably starts a new sentence), l. 210 and l. 247 missing space before opening bracket, l. 415 unnecessary hyphenation, l. 420 missing “tissue” in the 1st sentence.
- Shouldn't the ethics committee statement details be presented in a separate paragraph instead of within the acknowledgments? Check the guidelines.
- The description of the learning phases is still misleading to some extent – if the expert corrects the outcomes of the previous learning phase (caption of Fig. 2) are those not-corrected (i.e. from the first and second stage) also used to train the system (“Finally, we used all samples in the dataset for model training.”)?
- l. 289 the said image resolutions are DPIs?
- Category/class is used interchangeably – consider unifying.
- Maybe an answer to the question indicated in l. 370 might be answered by reverting back to the original ResNet architecture? Was such an experiment considered? Or it still requires more data to cover the variability of class 2 tissue.
- To check if image resolution is the main cause of texture flatness two images in the same lightning conditions of the same wound could be made. Maybe the main cause is not image resolution but conditions of image acquisition? Of course, if the input to the network is always 32x32 the perceived texture is dependent on image DPI, though.
Author Response
Reviewer Comments for Author(s)
Reviewer 2
The manuscript has been improved, however there are still language and editorial issues. Moreover there are still some statements that might be confusing to the reader. As previously, only three cases are presented in detail, therefore the outcomes can be considered only qualitative and robustness of the system for its daily application is not confirmed.
Specific issues:
- Proofreading by a native speaker is still recommended – there are still sentences hard to follow and/or language issues (e.g. l. 48, l. 58+, l. 97+, l. 101+, l. 121+, l. 157+, l. 173+, l. 208+, l. 362+).
Ans: This manuscript has been rewrite and proofreading by a native speaker.
- Inconsistent/incorrect capitalization, e.g. l. 75 (Clot), l. 120 (Biofilm), l.316 (Then This).
Ans: It has been corrected.
- Editorial: l.97 (not introduced acronym 3C), l. 163 (period instead of comma, “Photo” probably starts a new sentence), l. 210 and l. 247 missing space before opening bracket, l. 415 unnecessary hyphenation, l. 420 missing “tissue” in the 1st sentence.
Ans: It has been corrected.
- Shouldn't the ethics committee statement details be presented in a separate paragraph instead of within the acknowledgments? Check the guidelines.
Ans: We had separated paragraph of “Institutional Review Board Statement” as the following
“All subjects gave their informed consent for inclusion before they participated in the study. The study was conducted in accordance with the Declaration of Helsinki, and the protocol was approved by the Ethics Committee of by the institutional review board from university hospitals (IRB number: 201802012A3D001 and 201912056RINA).”
- The description of the learning phases is still misleading to some extent – if the expert corrects the outcomes of the previous learning phase (caption of Fig. 2) are those not-corrected (i.e. from the first and second stage) also used to train the system (“Finally, we used all samples in the dataset for model training.”)?
Ans: The experts can only modify the outcomes of the validation data (not the training data) and then insert them into the dataset as new samples. The new samples and the existing samples, from the first and the second rounds, are all correct samples; therefore, they were all used to train the model in the final stage.
- l. 289 the said image resolutions are DPIs?
Ans: Yes, we have added “dpi” into the wording.
- Category/class is used interchangeably – consider unifying.
Ans: We have unified this term to “class”.
- Maybe an answer to the question indicated in l. 370 might be answered by reverting back to the original ResNet architecture? Was such an experiment considered? Or it still requires more data to cover the variability of class 2 tissue.
Ans: Thank you for your suggestion. We tuned our model within a certain scaling range and found that the proposed architecture was best suited for the data size of our current stage. Of course, as we collect more wound image data, we will repeat the search for a perfect model size.
- To check if image resolution is the main cause of texture flatness two images in the same lightning conditions of the same wound could be made. Maybe the main cause is not image resolution but conditions of image acquisition? Of course, if the input to the network is always 32x32 the perceived texture is dependent on image DPI, though.
Ans: Thank you for your suggestion. We used gamma transformation for data augmentation, which may simulate different lightning conditions. This is why we suspect that image resolution is the main cause for this problem, and why we propose to resize all images to a consistent resolution in the future research.